# Bringing Clarity to Complex Pediatric Perioperative Care: Development of a Novel Pathway to Support Goal-Directed, Patient-Centered Care

**DOI:** 10.3390/cancers17091452

**Published:** 2025-04-26

**Authors:** Tiffany Lin, Neveada Raventhiranathan, Myra Ahmad, Andrew L. Feit, Allison E. Hotze, Grace L. Ker, Robert P. Moore, Sergio D. Bergese

**Affiliations:** 1Stony Brook University Renaissance School of Medicine, Stony Brook, NY 11794, USA; tiffany.lin@stonybrookmedicine.edu; 2Department of Anesthesiology, Stony Brook University Renaissance School of Medicine, Stony Brook, NY 11794, USA; neveada.raventhiranathan@stonybrookmedicine.edu (N.R.); andrew.feit@stonybrookmedicine.edu (A.L.F.); robert.moore5@stonybrookmedicine.edu (R.P.M.); 3Touro College of Osteopathic Medicine, New York, NY 10027, USA; myramahmad@gmail.com; 4Faculty of Graduate Studies in Advanced Practice Nursing, Stony Brook University School of Nursing, Stony Brook, NY 11794, USA; allison.hotze@stonybrookmedicine.edu; 5Department of Pediatrics, Stony Brook University Renaissance School of Medicine, Stony Brook, NY 11794, USA; grace.ker@stonybrookmedicine.edu

**Keywords:** pediatrics, pediatric surgery, pediatric oncology, pediatric palliative care

## Abstract

Advances in pediatric care have resulted in the ability to offer operative care that can result in outcomes that diverge from goals of care. We describe the development of a collaborative process that leverages palliative care consultation to ensure that complex surgical care remains family-centered and goal-directed.

## 1. Introduction

Advances in pediatric perioperative care have supported the safe provision of increasingly complex surgical care to patients with complicated disease and physiologic burdens [1]. Similarly, improvements in technology and care have altered the landscape of pediatric care, leading to a cohort of hospitalized children with increasingly complex disease burdens [2,3,4,5]. These parallel advances have led to unique challenges, including the ability to offer pediatric surgical interventions with an unclear risk–benefit profile.

In addition to the clinical and technical challenges created by the combination of complex disease burden and invasive surgical care, this evolving landscape may have a profound impact on informed consent and the ability to provide family-centered, goal-directed care. The inability to provide family-centered, goal-directed care can have devastating consequences, especially in the context of oncologic care that is typically rooted in thoughtful, shared decision-making. For example, consider the case of a teenage girl with newly diagnosed leukemia who is mechanically ventilated; she has a grim prognosis in the setting of ongoing intra-abdominal bleeding unresponsive to efforts by interventional radiology. The family was faced with a choice between supportive care and an exploratory laparotomy with an exceptionally high risk of intra-operative mortality. Making this decision would be a daunting task under ideal circumstances. A family may feel overwhelmed despite the outstanding efforts of the multitude of care givers involved in primary, intensive, consultative, and supportive care. In such a case, the ability to leverage an existing process designed to support complex decision-making that includes a skilled, neutral palliative care physician would prove invaluable.

Parental misunderstanding of the impact of surgery in the context of complex disease is an ongoing challenge [6] that is occurring with increasing frequency. For example, the majority of children receiving palliative care undergo at least one surgical intervention over the course of palliative treatment [7]. Surgeries may occur in the context of either symptomatic or end-of-life care. In general, parental misunderstanding is a clear challenge for all involved in pediatric perioperative care as there will always be a clear need to align family goals of care with the possible impact of invasive surgical interventions. The misalignment of care goals and clinical choices is a profound challenge, especially in light of the likely impact of surgery on quality of life.

In an effort to address these challenges and avoid the consequences of offering surgical care that may not align with family and patient goals of care, our institution assembled a collection of expert stakeholders to design a process supporting family decision-making related to perioperative care in the context of complex disease burden including malignancies. Of note, the process was intended to support choices that are ethically appropriate but not easily aligned with family goals or readily addressed by standard planning efforts. Ultimately, early and ongoing palliative care consultation was identified as a means to both articulate goals of care and provide a medically literate and neutral ombuds to support families and patients in the context of perioperative care planning.

This choice was rooted in both local experience and existing recommendations for pediatric palliative care that highlight the importance of focusing on shared objectives, recognizing cognitive biases, and practicing conflict resolution [8]. Our process was the product of a focused literature review and ongoing discussion that led to the development of a novel standardized approach to support families in the context of a potential surgical intervention that may not readily align with care goals.

This manuscript will provide an overview of our novel process and the existing literature supporting our choices.

## 2. Methods

A group of expert local stakeholders was identified and charged with developing a process to support patients and families with decision-making related to potential surgical interventions. The team had a stated goal of ensuring that decision-making was shared and goal-directed, leading to potential outcomes that aligned with stated care goals.

Team members included physicians, nurses, and nurse practitioners with expertise in pediatric anesthesia, pediatric intensive care, pediatric oncology, pediatric palliative care, pediatrics, ethics, pediatric surgery, and adult intensive care.

To support the development of a robust, evidence-based process, a systematic search of the current literature on pediatric and adult palliative care and surgery was performed. Searches in PubMed Central, EMBASE, CINAHL, Google Scholar, the Web of Science citation index, and the US clinical trials register were carried out using Boolean combinations of terms, including “pediatric,” “children,” “adult,” “palliative care,” “do not resuscitate,” and “surgery,” with the most recent update on 1 September 2024.

## 3. Shared Decision-Making

Shared decision-making is a key goal of family-centered pediatric care and a central tenant of pediatric palliative care [9,10,11]. Similarly, key palliative care goals include highlighting shared objectives, identifying cognitive biases, and facilitating conflict resolution [8].

The vital need to apply this approach to care and decision-making is reflected by the growth of the sub-specialty of pediatric palliative care which has occurred in parallel to the increasingly complex disease burdens encountered in inpatient pediatric care settings [2,3,4,5]. For example, in 2006, pediatric palliative care was formally recognized as a sub-specialty through a combined effort of the American Board of Medical Specialties, the American Board of Surgery, and the American Board of Pediatrics [11]. Since this initial establishment, there has been a proliferation of pediatric palliative care practices and training programs.

In light of both growing access to pediatric palliative care and the potential impact of the central principles of the practice to support shared decision-making, the inclusion of pediatric palliative care doctrines was identified as a key element of our new process. However, considering the overlap of pediatric palliative care principles with several existing aspects of pediatric perioperative care [12], a thoughtful review of the related literature was conducted. This review was viewed as vital to ensuring the efficient use of resources and to support the timely provision of goal-directed care. Palliative care in the adult perioperative setting, palliative care in the pediatric perioperative setting, surgical care at the end of life, and palliative care in the setting of pediatric malignancies were identified as key areas for review.

## 4. Palliative Care in the Adult Perioperative Setting

A small but growing body of literature supports the integration of palliative care principles into the care of adults with complex disease burden. For example, there is increasing support for the integration of palliative care principles into the provision of critical care [13,14]. Both a consultative and integrative model for critical care have been adopted in these settings. The consultative model suggests that an intensivist consults a trained palliative care specialist to manage the patient when conditions for the initiation of palliative care are met. The integrative model involves intensivists in concert with the entire ICU team assessing the range of palliative care interventions and providing them for patients and their families [13].

Similarly, several models have been developed to incorporate palliative care into the adult perioperative setting. The perioperative surgical home (PSH) is a care model that emphasizes both patient-centered care and shared decision-making throughout the perioperative period [15]. The adoption of the PSH model provides an opportunity to conduct discussions with patients, family members, and an interdisciplinary team consisting of not only surgery and palliative care physicians but also strongly connected allied professionals, including but not limited to psychiatrists, therapists, and social workers [15]. Despite a paucity of data, Cobert et al. advocate for greater adoption of this patient-centered process, including the engagement of palliative care consultants when needed, as it is a simple means to meet complex care needs [15].

The vital role of the principles of palliative care in adult medicine is also reflected by advocacy for an emphasis on education related to palliative care principles and enhanced communication skills. These goals have been identified as a likely unmet need in anesthesiology training [16,17].

Overall, the existing adult literature clearly supports a growing role for the thoughtful application of the principles of palliative care and expert palliative care consultation in the care of adults with complex disease burden, especially throughout the perioperative period.

## 5. Palliative Care in the Pediatric Perioperative Setting

While there is a growing body of literature related to palliative care in the adult perioperative setting, the pediatric literature is limited. This may be the product of epidemiological differences, the complexity of pediatric palliative needs, and the ethical difficulties that can surround end-of-life care in pediatric populations [6,18,19,20].

Despite the limited literature related to the perioperative period, there is a growing and robust body of literature supporting a role for pediatric palliative care consultation in diverse clinical settings. A prospective observational cohort study of pediatric palliative care patients in the United States and Canada showed that patients’ ages ranged from less than 1 month old to 19 years and older, underscoring the breadth and the range of the practice [21]. Consultation can occur in a wide variety of clinical settings, including congenital anomalies, genetic conditions, inborn errors of metabolism, skeletal disorders, neuromuscular disorders, and malignancies [21,22,23].

Surgical interventions for the diverse cohort of palliative care patients are becoming increasingly common. Recent cohort analyses have shown that most children receiving palliative care services undergo at least one surgical intervention, with the median value falling between four to thirteen interventions [7,24]. The time distribution of these interventions also varies depending on a pediatric patient’s diagnosis, with acute illnesses requiring more high-volume and high-risk interventions over a relatively short time period, while more chronic illnesses require a higher volume of individually lower-risk interventions performed over a prolonged course [7].

These data may underestimate the number of surgical interventions performed on children who would benefit from palliative care consultation. As pediatric palliative care consults remain underutilized, with less than half of consults occurring before a pediatric patient’s late-stage surgical procedure, and as many as 12% of consults occurring within one day of death, which may not provide sufficient time for patients and their families to meaningfully engage with palliative care resources [25]. In an effort to meet this recognized shortcoming, there have been proposed models and programs to integrate pediatric palliative care in various clinical settings [12]. However, widespread use may be limited by the difficulty of implementing these models within the financial and logistical constraints of hospital systems [12].

While limited, the literature demonstrates the benefits of perioperative palliative care. Consultation supports the effective management of pain and anxiety that may accompany surgical interventions and can reduce the intensity and quantity of medical interventions [26]. Early involvement and thorough integration of pediatric palliative care results in better-elaborated hospice discussions with appropriate time for end-of-life planning, earlier documentation of do-not-resuscitate (DNR) orders, less suffering as reported by both patients and parents, and fewer in-hospital deaths [27]. Overall, the literature is quite limited but clearly suggestive of the need to thoughtfully expand perioperative access to pediatric palliative care.

## 6. Palliative Care and Pediatric Oncology

The care of pediatric oncology patients can provide clear insight into the impact of expanded access to palliative care. Palliative care services for a pediatric oncology patient can be offered at any point in the disease trajectory, even when a patient may seem “well.” These services can aid the management of complex clinical symptoms, provide family and individual support, and promote advanced care planning throughout a patient’s disease and treatment course [28]. Pediatric palliation for oncology cases often requires coordination with ongoing medical treatments such as chemotherapy sessions, radiation therapy, or stem cell transplantation, all of which involve open and clear communication between palliative care, various medical teams, social work services, and patients and families [29]. Such thoughtful coordination could clearly benefit patients with complex disease throughout the perioperative period.

Pediatric oncology patients will often report a range of symptoms that negatively affect their quality of life, with some of the most prevalent symptoms including pain, fatigue, drowsiness, and irritability [30]. During a patient’s last 12 weeks of life, studies have shown that patients’ pain prevalence often increases [31]. Families and caregivers may report that patients show a lack of appetite, lack of energy, or low mobility [32]. It is rare for pediatric palliative care patients to only report one or even two primary symptoms; research has found that patients and their caregivers commonly report six or seven symptoms that require management and attention [33]. Palliative care can be critical in addressing pediatric oncology patients’ high symptom burden and improving their comfort throughout a prolonged illness course. Specialized pediatric palliative care has been shown to provide patients with better symptom control, often involving less pain and reduced dyspnea [26]. Additionally, care can lead to fewer surgical procedures and intensive care stays [26]. The impact of palliative care in these settings would suggest benefit throughout the perioperative care of children with complex baseline disease burden.

The care of children with cancer may also provide insight into possible fiscal impact of the perioperative care of children with complex disease burden, including the additional impact of palliative care consultation during the perioperative period. The financial costs of treatment significantly impact individual families and healthcare. A meta-analysis examining the costs of cancer treatment for children and adolescents in various countries found in the first year of cancer diagnosis for children that the mean healthcare cost was over USD 130,000 in Canada and USD 61,855 for adolescents in Canada [34]. Australia presented a similar pattern with cancer diagnosis in children costing a median annual of USD 86,799 and adolescents costing USD 47,487. For children and adolescents diagnosed with cancer, the continuing cost of care for treatment was in a higher range for children in comparison to adolescents [34]. In Canada, hospitalization cost for continuing care was approximately USD 16,253 for children and USD 7749 for adolescents [34]. The costs tended to be highest in the last twelve months of life, which are referred to as the end-of-life care phase; the cost for children in Canada reached over USD 300,000 and for adolescents approximately USD 235,2653. The largest factor of these healthcare costs was the cost of hospital admission, while end-of-life and palliative care contributed a smaller component of the overall healthcare cost [34].

Overall, this would suggest a similar limited impact of the additional cost of perioperative palliative care consultation for children with complex disease burden. Additionally, given the high cost associated with hospitalization [34] and the potential for palliative care to reduce such costly perioperative events as intensive care admission [26], consultation may actually lead to reduced overall costs.

## 7. Novel Pediatric Perioperative Process

In light of the literature review and local clinical strengths supporting benefit for perioperative palliative care consultation, we developed a novel perioperative care pathway for children with complex disease burden presenting for surgery with an unclear risk–benefit profile. The effort leverages palliative care consultation to allow for the clear identification of patient and family goals to ensure goal-directed perioperative care in the context of an ambiguous risk–benefit profile.

The ability of early palliative care engagement to meet this unique care need is reflected by the American Academy of Pediatrics (AAP) recommendation of connecting palliative care programs and team members with a child’s usual medical caregivers—ranging from pediatricians, family medicine doctors, surgeons, and so forth—to provide fluid, comprehensive, and holistic care [16]. Essentially, the AAP suggests that engagement of palliative care in all aspects of management would benefit patients with complex illnesses. In fact, the AAP specifically recommends that individualized, family-centered palliative care be offered at the time of diagnosis and continued throughout a patient’s illness course.

In light of the existing literature and AAP recommendations, we have developed a novel integrative process to ensure that perioperative planning aligns with family and patient care goals that is outlined below (Figure 1 and Figure 2).

1. **Identification of Patients**: Patients with complex disease burden that may require surgery with an unclear risk–benefit profile are identified for possible inclusion into the pathway at the time of diagnosis or at the time of perioperative care planning. As this process is viewed as a key effort to ensure that care is aligned with patient and family goals, anyone involved in primary care, consultative care, or perioperative planning can identify and refer a patient.

2. **Palliative Care Consultation:** Consultation is obtained with the specific objectives of articulating goals of care and establishing a relationship that can continue longitudinally.

3. **Multidisciplinary Team Meeting(s):** When a possible surgical intervention is considered, a multidisciplinary team meeting is convened, including the palliative care team, the primary pediatric team, pediatric surgical consultants, pediatric anesthesia consultants, and any consultants engaged in the ongoing management of the child. This meeting allows for the articulation of care goals and a nuanced discussion of the potential of perioperative care choices to impact and align with patient-centered care goals.

4. **Family Meeting:** In a similar manner, a family meeting is convened involving the consultants engaged in ongoing care and possible perioperative care. The impact of potential surgical care and the alignment of these changes with known goals of care is discussed. The palliative care consultant is uniquely positioned to support the family as a knowledgeable and neutral ombuds. This meeting should allow for thoughtful decision-making in the context of challenging perioperative choices.

5. **Continuity of Care:** All members of the care team remain engaged throughout the perioperative process and beyond. This ensures that care will remain patient-centered and goal-directed. Any of the prior steps of this process may be repeated as needed to support this goal.

This process was initially applied to pediatric oncology patients due to both the complexity of care goals and existing relationships with our pediatric palliative care team. The process was very well received and subsequently applied throughout the children’s hospital, including for the care of neonates, antenatal care planning, and the care of critically ill children.

For example, consider the case of the family of a syndromic neonate with a grim prognosis and a presumed very limited life span that simply wish to spend as much time as possible at home with their child. Several pathways could accomplish this goal, including discharge with home hospice or discharge following more invasive supportive measures, including placement of a tracheostomy and a feeding tube. Engagement in our supportive decision-making process would be vital to ensuring that specific choices align with broad family goals. Both pathways could accomplish the same goal but carry unique and challenging risk and benefit profiles.

## 8. Summary

The alignment of possible pediatric surgical outcomes and patient care goals can be a challenge in the context of significant baseline disease burden. We have developed a process that leverages pediatric palliative care consultation in concert with multidisciplinary care planning to ensure that perioperative care is goal-directed and patient-centered for all patients, including those with substantial, existing physiologic challenges. The pathway has been well-received and expanded hospital-wide following initial application for oncology patients.

Our pathway has the potential to promote improved communication, support the enhanced ability to direct care from the perspective of family goals, improve analgesia, lead to reduced morbidity, and reduce costs. Additionally, ongoing multidisciplinary interaction will serve as an educational template to assist the next generation of consultants with mastery of challenging, nuanced, and collaborative perioperative communication. Further data are needed to fully understand the impact of this process. Similar processes should be considered at other institutions.

## 9. Conclusions

Despite the typical outstanding supportive efforts of numerous professionals, complex physiology combined with challenging surgical interventions can lead to the misalignment of care choices and known goals of care. We describe a novel multidisciplinary process designed to thoughtfully resolve this vital care gap.

## Figures and Tables

**Figure 1 cancers-17-01452-f001:**
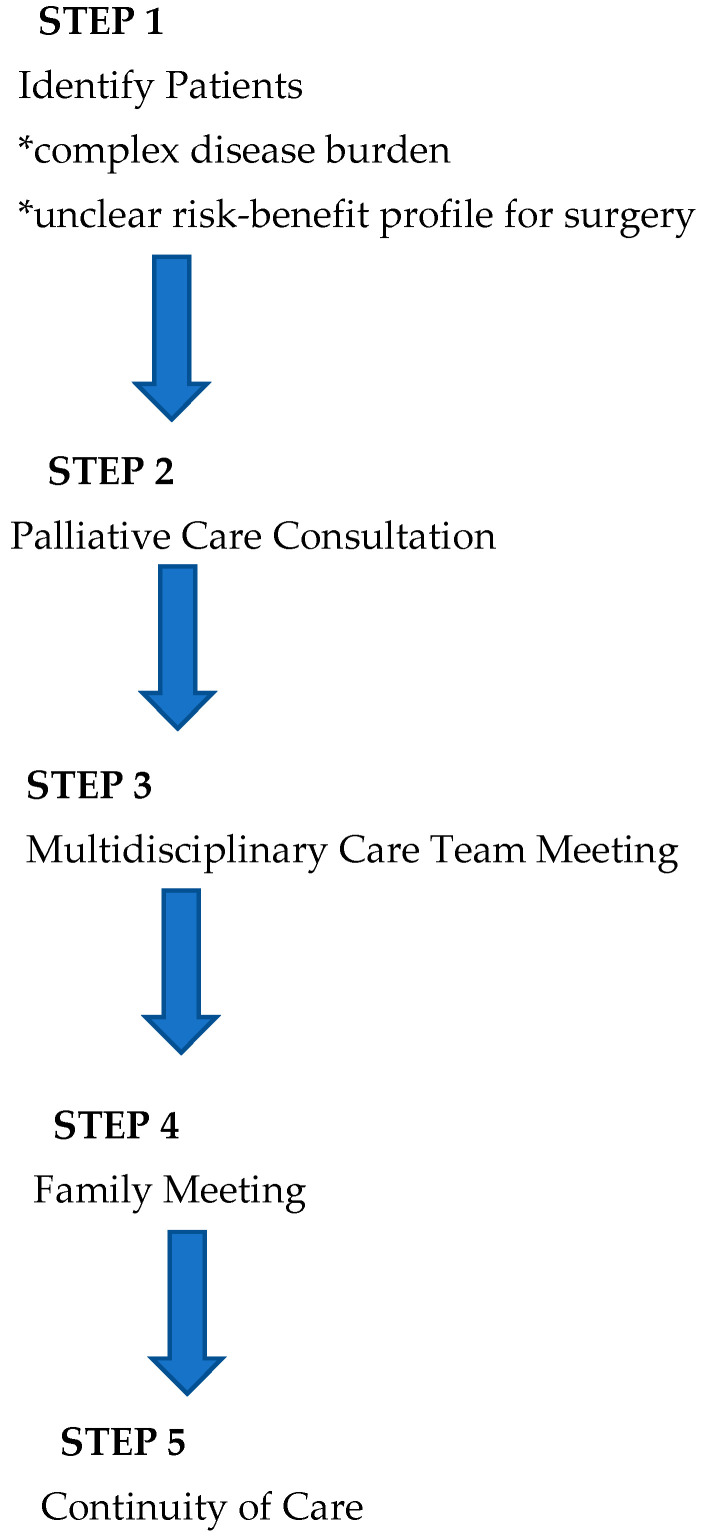
Overview of process.

**Figure 2 cancers-17-01452-f002:**
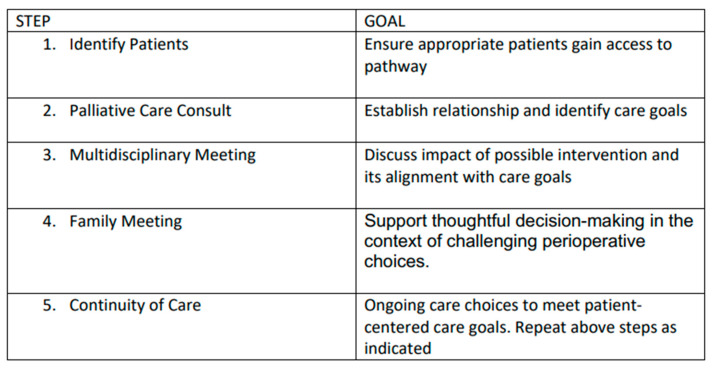
Overview of goals of each step of our novel process.

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
