# Peer review of "Bringing Clarity to Complex Pediatric Perioperative Care: Development of a Novel Pathway to Support Goal-Directed, Patient-Centered Care"

_cancers, 2025, doi:10.3390/cancers17091452_

Round 1

Reviewer 1 Report

Comments and Suggestions for Authors

In this article, the authors briefly review the complex decision-making aspects of palliative surgery in adults and children and present a perioperative care pathway for children with complex disease burden and unclear risk-benefit profile in use at their institution. The problem is current and of interest to readers.

I recommend a shorter title and a more concise style in general. It would be useful to go into more detail, for example by indicating the period in which the perioperative care pathway was adopted, which were the most frequent situations in which the multidisciplinary team evaluated pediatric oncology patients, and for which type of interventions.

Author Response

We refer to the above paper and would like to thank you again for your kind words and suggestions. We have worked towards amending our manuscript accordingly and are pleased to submit our revised manuscript for your consideration. Changes have been highlighted in yellow.

Reviewer 1:

Reviewer 1 Comment: In this article, the authors briefly review the complex decision-making aspects of palliative surgery in adults and children and present a perioperative care pathway for children with complex disease burden and unclear risk-benefit profile in use at their institution. The problem is current and of interest to readers.

Author response:  Thank you for the thoughtful comments.

Reviewer 1 Comment: I recommend a shorter title and a more concise style in general.

Author response: Thank you -  the title has been revised and the body has been edited for clarity.

Reviewer 1 Comment: It would be useful to go into more detail, for example by indicating the period in which the perioperative care pathway was adopted, which were the most frequent situations in which the multidisciplinary team evaluated pediatric oncology patients, and for which type of interventions.

Author response: Thank you for the comment. The pathway has been in use for around 18-months. While it has been well received and impactful, we plan to wait for a larger cohort of cases before sharing data.

Thank you for your time and expertise.

Reviewer 2 Report

Comments and Suggestions for Authors

The manuscript by Lin et al reviews the current literature as regards to approaching palliative care in a pediatric oncology context. This review is well written and informative.

I have two suggestions: 

  1. In the section that describes the novel process at work at the authors' institution, the first step was to "Identify Patients". Please describe how the team did this. Which patients? why? in what contexts?
  2. Again in this section, providing an example of how this process worked would be informative to the readers.

Author Response

We refer to the above paper and would like to thank you again for your kind words and suggestions. We have worked towards amending our manuscript accordingly and are pleased to submit our revised manuscript for your consideration. Changes have been highlighted in yellow.

Reviewer 2:

Reviewer 2 Comment: The manuscript by Lin et al reviews the current literature as regards to approaching palliative care in a pediatric oncology context. This review is well written and informative. I have two suggestions:

Comment 1: In the section that describes the novel process at work at the authors' institution, the first step was to "Identify Patients". Please describe how the team did this. Which patients? why? in what contexts?

Author response: Thank you for the comment. The revised manuscript more clearly describes this aspect of the process. The intention of the effort is to ensure the care outcomes and family goals remain aligned. Accordingly, we have a liberal inclusion process. Any care team member can initiate the pathway.

Reviewer 2 Comment: Again in this section, providing an example of how this process worked would be informative to the readers.

Author response: Thank you for the suggestion – the revised document provides added clarity related to the application of the process.

  Thank you for your time and expertise.

Reviewer 3 Report

Comments and Suggestions for Authors

Intro:

  • “This can have devastating consequences, especially in the context of oncologic care that is typically rooted in thoughtful shared decision making.” The mention of oncology caught me off guard here as the paper seemed to be staying generalized in terms of subspecialty until this sentence. Could this not be true for PICU patients or other complicated life limiting diseases such as organ transplant or cystic fibrosis? It would help the reader to have an example here such as:

“This can have devastating consequences, especially in the context of oncologic care where parents may be asked to provide consent for surgical procedures without understanding that prognostic outcomes have shifted significantly.”

  • Parental misunderstanding of the impact of surgery in the context of complex disease 44 is an ongoing challenge [6] that is occurring with increasing frequency. For example, the 45 majority of children receiving palliative care undergo at least one surgical intervention 46 over the course of palliative treatment [7].
    • This is another good point but again, given the wide readership, it would be good to distinguish children receiving palliative care in the context of symptom palliation vs children who are the end of life. Palliative can comprise both and it would benefit to understand what the point of the sentence is.

The team seems to have very little understanding of the need for and benefits of pediatric psychologists in helping intensivists understand the lived experiences and perspectives of children and their caregivers. The use of “psychiatrists, therapists, and social workers” without mention of pediatric/child psychology is sort of egregious.

The authors being all from one institution should consider including their psychology colleagues in review of this protocol because this paper reads as though it’s written by a bunch of anesthesiologists who are trying to think through concrete protocols.

There should be way more mention of psychology, ethics, and social work. Likewise, there should be early and frequent mention of child life specialists. The authors frankly seem to be adult providers writing a pediatric paper.

If this is a paper on pediatrics, there should be much less on adult models. Adult health centers have less robust teams that include the specialists above. Also, in pediatrics, there is the ability for concurrent care that always families to not have to choose between disease directed care and palliation, something that is not available to most adults due to insurance limits.

Most children being considered for surgery in these settings are cared for by hospitalists and pediatric critical care doctors, both of which are well versed in understanding how to rope in social work, psychology, ethics and to hold family meetings that enable shared understanding. The paper is written like authors are not aware of this.

The novel perioperative process is not a bad one but it reads as though surgeons and anestiologists act as primary teams in hospitals, when this is often not the case outside of patients admitted for primary orthopedic procedures. So not sure this is needed or the context needs to be clarified.

Author Response

We refer to the above paper and would like to thank you again for your kind words and suggestions. We have worked towards amending our manuscript accordingly and are pleased to submit our revised manuscript for your consideration. Changes have been highlighted in yellow.  

Reviewer 3 Comment: Intro - “This can have devastating consequences, especially in the context of oncologic care that is typically rooted in thoughtful shared decision making.” The mention of oncology caught me off guard here as the paper seemed to be staying generalized in terms of subspecialty until this sentence. Could this not be true for PICU patients or other complicated life limiting diseases such as organ transplant or cystic fibrosis? It would help the reader to have an example here such as: “This can have devastating consequences, especially in the context of oncologic care where parents may be asked to provide consent for surgical procedures without understanding that prognostic outcomes have shifted significantly.”

Author response: The phrase is meant to provide a broad example of a type of patient who may be vulnerable to consenting to surgical care that does not clearly align with care goals. It is intended to provide a general point of reference for a reader. We have further clarified this statement with the addition of a more specific example.

Reviewer 3 Comment: “Parental misunderstanding of the impact of surgery in the context of complex disease 44 is an ongoing challenge [6] that is occurring with increasing frequency. For example, the 45 majority of children receiving palliative care undergo at least one surgical intervention 46 over the course of palliative treatment [7].” This is another good point but again, given the wide readership, it would be good to distinguish children receiving palliative care in the context of symptom palliation vs children who are the end of life. Palliative can comprise both and it would benefit to understand what the point of the sentence is.

Author response: In light of the included references, it is implicit that all children who present for palliative would be included in this cohort. This is now explicitly stated. Thank you for the suggestion.

Reviewer 3 Comment: The team seems to have very little understanding of the need for and benefits of pediatric psychologists in helping intensivists understand the lived experiences and perspectives of children and their caregivers. The use of “psychiatrists, therapists, and social workers” without mention of pediatric/child psychology is sort of egregious. The authors being all from one institution should consider including their psychology colleagues in review of this protocol because this paper reads as though it’s written by a bunch of anesthesiologists who are trying to think through concrete protocols. There should be way more mention of psychology, ethics, and social work. Likewise, there should be early and frequent mention of child life specialists.

Author response: We would absolutely agree that ignoring the work of the multitude of excellent professionals engaged in complex pediatric care would be inappropriate. However, this is an incorrect assertion about this manuscript.

We describe a process that is intended to bridge gaps in complex consultative care that occur rarely but can have a profound impact. We have more directly and clearly reinforced this aspect of the effort. Accordingly, it is assumed that patients are already benefitting from outstanding standard multidisciplinary care at the time of pathway initiation.

The authors include tertiary care pediatricians, pediatric intensivists, pediatric anesthesiologists and primary care pediatric providers. We are always grateful and appreciative of all our colleagues. However, providing a complete list of all our wonderful pediatric fellow travelers would be impractical and somewhat precedent setting for a paper focused on efforts directly related to support sub-specialty care.

Thank you for the comments that helped remove any uncertainty about his effort while allowing us to clearly communicate our profound respect for all our colleagues.

Reviewer 3 Comment: The authors frankly seem to be adult providers writing a pediatric paper. If this is a paper on pediatrics, there should be much less on adult models. Adult health centers have less robust teams that include the specialists above. Also, in pediatrics, there is the ability for concurrent care that always families to not have to choose between disease directed care and palliation, something that is not available to most adults due to insurance limits. Most children being considered for surgery in these settings are cared for by hospitalists and pediatric critical care doctors, both of which are well versed in understanding how to rope in social work, psychology, ethics and to hold family meetings that enable shared understanding. The paper is written like authors are not aware of this.

The novel perioperative process is not a bad one but it reads as though surgeons and anestiologists act as primary teams in hospitals, when this is often not the case outside of patients admitted for primary orthopedic procedures. So not sure this is needed or the context needs to be clarified.

Author response:

Thank you for the comments. As noted previously, this effort is aimed to bridge a rare gap in the typical outstanding pediatric standard of care. By definition, the gap that we identified would suggest that these typically excellent efforts are not always successful. Accordingly, a unique effort was needed and created. It was decided that a thoughtful novel effort should be rooted in the totality of applicable evidence including adult efforts. This is a common approach for pediatric care pathway development including such successful interventions as enhanced recovery after surgery (ERAS) guidelines.

Our effort and manuscript are certainly not intended to criticize or diminish outstanding standard efforts, resources and professionals. The effort is an effort to improve care when the limits of standard care have been exceeded. Interpreting the paper as not understanding impactful standard care would be akin to viewing a description of a novel surgical approach as an effort to ignore the extraordinary efforts of numerous professionals involved in both routine and non-operative care. This was certainly not our intention and we are grateful that you were able to highlight and exemplify the impact of this view of the manuscript.

The revised document explicitly describes the specific aim of the effort and acknowledges the typical success and collaborative nature of routine multi-disciplinary care.

Again, as noted previously, the authors include tertiary care pediatricians, pediatric intensivists, pediatric anesthesiologists and primary care pediatric providers. Our group is heavily engaged nationally as clinicians and academicians.  The list also includes two adult specialists with expertise in ethics and complex care planning. One of these authors is widely recognized as a scholar of international repute. No one in this list would diminish the mutli-disciplinary efforts that are the basis of all pediatric care. We again are grateful that you were able to highlight that readers could have a subtle but concerning misunderstanding.

The revised document clarifies the consultative nature of the effort and that is intended as an extension and not an effort to override excellent standard care.

  Thank you for your time and expertise.